# Bipolar Radiofrequency Ablation of Painful Spinal Bone Metastases Performed under Local Anesthesia: Feasibility Regarding Patient’s Experience and Pain Outcome

**DOI:** 10.3390/medicina57090966

**Published:** 2021-09-15

**Authors:** Adrian Kastler, Daniel-Ange Barbé, Guillaume Alemann, Georges Hadjidekov, Francois H. Cornelis, Bruno Kastler

**Affiliations:** 1Diagnostic and Interventional Neuroradiology Unit, CHUGA Grenoble Hospital, Université Grenoble Alpes, 38400 Saint Martin d’Heres, France; 2Imagerie Medicale Pays D’agde, 34300 Agde, France; danielange.barbe@gmail.com; 3Cabinet Médial Clemenceau Selestat, 67600 Selestat, France; guillaume.alemann@gmail.com; 4Department of Radiology, University Hospital Lozenets, 1407 Sofia, Bulgaria; jordiman76@yahoo.com; 5Adult Radiology Department, Necker Hospital, Paris V University, 75015 Paris, France; cornelisfrancois@gmail.com; 6Department of Radiology, Memorial Sloan Kettering Cancer Center, New York, NY 10065, USA; radioakastler@gmail.com

**Keywords:** radiofrequency ablation, vertebroplasty, CT guidance, metastasis, local anesthesia, palliative care

## Abstract

*Background and objectives:* To assess the pain relief of bipolar RFA combined or not with vertebroplasty in patients with painful vertebral metastases and to evaluate the feasibility and tolerance of the RFA procedure performed under local anesthesia. *Materials and Methods*: 25 patients (18 men, 7 women, mean age: 60.X y.o) with refractory painful vertebral metastasis were consecutively included between 2012 and 2019. A total of 29 radiofrequency ablation (RFA) procedures were performed under CT guidance, local anesthesia and nitrous oxide inhalation, including 16 procedures combined with vertebroplasty for bone consolidation purposes. Pain efficacy was clinically evaluated using the visual analogue scale (VAS) at day 1, 1 month, 3 months, 6 months and 12 months, and the tolerance of the procedure was evaluated. *Results:* Procedure tolerance was graded as either not painful or tolerable in 97% of cases. Follow-up postprocedure mean VAS score decrease was 74% at day 1: 6.6 (*p* < 0.001), 79% at 1 month: 6.6 (*p* < 0.001), 79% at 3 months: 6.5 (*p* < 0.001), 77% at 6 months, and 79% at 12 months: 6.6 (*p* < 0.001). *Conclusions:* Bipolar RFA, with or without combined vertebroplasty, appears to be an effective and reliable technique for the treatment of refractory vertebral metastases in patients in the palliative care setting. It is a feasible procedure under local anesthesia which is well tolerated by patients therefore allowing to broaden the indications of such procedures. Field of study: interventional radiology.

## 1. Introduction

Bone is one of the most common metastatic sites, and 50% of pain experienced by cancer patients originates from bone metastases [1,2,3,4]. A spinal location may represent up to 80% of bone metastases [1]. In case of an extension to the neural structures, pain can be both radicular (exaggerated by percussion or palpation) and/or mechanical (exacerbated by movement) [5,6]. At advanced stages of the disease, pain may become intolerable, and refractory to conventional therapies causing walking disabilities, psychological and functional impact can thus impair markedly the quality of live [2,3,4,7].

Due to the short life expectancy of affected patients, treatment regimens are most often palliative rather than curative. Therefore, quick pain relief has become a priority in these patients suffering from refractory bone pain.

Several conventional treatment options have been described including opioids, hormone therapy, chemotherapy, radiotherapy and surgery, which all have side effects and contraindications. Radiotherapy remains the gold standard, but up to 20% of patients are nonresponders, and the reported maximum benefit is obtained with a delay of 5–20 weeks after completion of treatment [8,9,10].

In these two past decades, interventional percutaneous image-guided techniques have emerged with satisfactory results in the management of vertebral metastasis such as vertebroplasty [11,12], radiofrequency ablation and microwave ablation [13,14,15], combined radiofrequency and cementoplasty [13,16] 16 or cryotherapy [17]. Bone percutaneous ablation is a mini-invasive treatment which presents several advantages, and it is a repeatable treatment with no limitation doses from the skin exposure compared to radiotherapy and remains a treatment with no interference with systemic treatments, especially those delaying healing, contrary to open-surgery procedures.

Most studies to date have assessed the effectiveness and safety of RFA with the use RF monopolar systems (which require grounding pads with a potential risk of skin burn) and were mainly performed either under general anesthesia or conscious sedation [18,19,20].

The objective of this study was to assess the pain relief of bipolar RFA combined or not with vertebroplasty in patients with painful vertebral metastases and to evaluate the feasibility and tolerance of the RFA procedures performed under local anesthesia and nitrous oxide ventilation.

## 2. Materials and Methods

Twenty-five consecutive patients (18 men, 7 women, mean age: 60.X y.o) were recruited between 2012 and 2019. All included patients presented painful spinal metastases refractory to all previously attempted conventional therapies, including opioids and radiotherapy, and the decision to undergo ablation was decided in a multidisciplinary meeting. The decision to perform adjunct vertebroplasty was made based on the location, type and extent of the lesion, and the Kostuik score [21] was used to predict fracture risk. In case of pathological fractures, cement injection was performed. Patient and lesion characteristics are detailed in Table 1. The average volume of treated tumors was 10.5 mL. The mean Karnofsky performance status was 59 (range 40–80).

Exclusion criteria were as follows: patients with tolerable pain (VAS < 5), locoregional or systemic infection at the time of inclusion, coagulation disorders.

### 2.1. Procedure and Anesthesia

Procedures were performed during a short hospital stay. CT guidance was used for targeting lesions.

Strict local anesthetic protocol was followed as previsoulsy described [22] and included:Needle pathway local anesthetic injection (a mixture of fast- and slow-acting anesthetic (lidocaine hydrochloride 1% (1/3) and ropivacaine hydrochloride 0.25% (2/3): from skin entry point to tumor, associated to an intratumoral block with the same mixture. Quantities of injectant depended on both tumor size and patient’s tolerance, without exceeding dose limits [10].Inhalation of nitrous oxide throughout the procedure.IV administration of paracetamol (1 g) started 5 min prior to procedure. IV injection of nalbuphin (20 mg) could be added on demand in case of persisting pain.

### 2.2. RFA Technique

Strict aseptic technique was assured during the procedure. From one to two bipolar RF needles (17 G diameter exposed tip, 20–30 mm, Celon Prosurge, Teltow, Germany) were introduced coaxially through a 13 gauge biopsy needle (t’CD II, Thiebaud, Thonon, France). Technical success was defined as the ability to successfully place the RFA probe in the center of the lesion in case of a unique lesion, and at least 1.5 cm distance between probes in case of several needles, and to perform RFA (Figure 1 and Figure 2). Duration and power of RFA depended on tumor size, manufacturer’s recommendation, patient’s tolerance and our own experience.

Multiple RFA cycles and/or multiple needle approaches were performed for large lesions (>4 cm) (see Table 1). In case of adjunct cementoplasty, cement injection was performed through the same needle after RFA probe retrieval, and an average of 4 mL of acrylic cement was injected. In case of proximity to a neural structure, a thermocouple was used to monitor intraprocedural temperature.

### 2.3. Pain Assessment

Visual analogue scale scoring was used to assess procedure effectiveness: before procedure, at day 1, 1st month, 3rd month, 6th month, and 12th month. A 50% pain decrease at one month on VAS was considered as a positive response. Intraprocedural pain and tolerance to the procedure was measured on a 0–2 scale: 0 = no pain, 1 = tolerable pain, 2 = intolerable pain.

## 3. Statistical Analysis

Continuous variables were expressed as mean +/− SD. The Shapiro–Wilk test was used to determine whether variables came from a normally distributed population. A Friedmann’s variance analysis and Student’s t test on paired samples were performed between the groups. A value of *p* < 0.05 was considered significant.

## 4. Results

### 4.1. Procedure

The technical success rate was 100%. The average duration RFA procedure was 23 +/− 9.9 min. A thermocouple was used in 17 cases, 12 cases of radicular nerve monitoring, and 5 cases of epidural monitoring. Multiples needles were performed in: seven cases with two needles and two cases with three needles. No clinical complications were noted during or after procedure. Analysis of postprocedural CT did not reveal the immediate procedure related complications.

Vertebroplasty was performed in seven cases of pathological fracture and nine cases of fracture prevention. In these cases of vertebroplasty, 11 of 16 postprocedural CT showed minor cement leakage with no clinical expression (prevertebral, epidural or intradiscal).

Constant oral contact was made with the patient during the whole procedure especially during the ablation phase, and the patient was told to alert, in case of pain radiation in the legs. Moreover, every minute, sensitive and motor testing was performed in order to ensure the lack of neurological damage.

### 4.2. Pain

The per procedure tolerance was rated as follows: 0 for 16 procedures (55%, 16/29), 1 for 12 (41%, 12/29) and 2 for 1 procedure (3%, 1/29).

One patient was lost to follow up at 12 months. Mean VAS before the procedure was 8.4/10. A significant reduction in pain was obtained in 24/29 (83%) procedures at 1 month. Details are summarized in Table 1 and Table 2 and illustrated in Figure 3.

Therapeutic failures on pain palliation were observed in three procedures with an advanced stage of disease. In each case, lesions were very large with significant prevertebral soft tissue and foraminal invasion. In five other patients, a pain decrease of less than 50% was noted: at 1 month (one patient), at 3 months (two patients), at 6 months (two patients) and at 1 year (two patients). Significant tumor lesion growth or new lesions were observed in these patients.

## 5. Discussion

Many therapeutic options are available for patient suffering from spinal metastasis.. Specific medication may lead to well-known side effects, gastrointestinal (NSAIDs, corticosteroids and opioids), neurological (morphine-impaired consciousness, etc.) or intolerance. Chemotherapy has systemic toxic effects and a delayed action. External beam radiotherapy remains the treatment of choice for the palliative treatment of metastatic bone tumors but is not effective in 20–30% cases (10 patients in our series). Furthermore, its analgesic effect is delayed 12–20 weeks [8]. Surgery is often not appropriate in late-stage cancer patient as it remains very invasive with a long recovery. Percutaneous RFA treatment offers a useful alternative for patients in palliative care units as pain improvement occurs very rapidly after treatment, as shown by several previous studies which have reported the usefulness, safety and rapid pain release after RFA for spinal lesions [19,20,23,24]. The choice of bipolar radiofrequency was made on the well-known advantages of bipolar RFA as opposed to monopolar RF-better control of the ablation zone [25] and no risk of skin burning due to grounding pads not contraindicated in pacemakers wearers [13,16]. Moreover, microwave ablation, although it has been reported to be feasible in vertebral lesions [14], presents higher risks of complications [26].

The results of our study are consistent with those of the recent literature [27]. Indeed, 82% of patients had an analgesic satisfying results at 1 month with significant long term pain palliation. Pain alleviation was obtained immediately after procedure and ensured significant lasting pain relief. Thus, end-of-life quality was improved in these patients suffering from intractable pain. The recovery of walking was possible for all patients (helping prevent decubitus complications, which are not rare). These results were obtained with a minimally invasive procedure under local anesthesia and nitrous oxide ventilation, which was very well tolerated by the patients as >90% of the patients graded the procedure as either nonpainful or tolerable. These findings are possible because careful attention was made to follow a well-codified anesthetic/pain management protocol which was strictly applied for each patient. Previous studies on RFA in the bone metastases report either the use of general anesthesia or conscious sedation [18,28]. However, a previous study assessing radiofrequency of extra axial metastasis has shown the feasibility of RF procedure under local anesthesia alone [13]. Local anesthesia has several advantages over general anesthesia: it allows per procedural clinical monitoring which helps to evaluate procedure tolerance and allows detection of possible neurological complications; this advantage is major, as spine thermal ablation may result in neurological impairment [29], and while previous authors have advised the use protective measures with good results [30,31], authors have already reported the use of local anesthesia and clinical intraprocedural assessment with microwave ablations, with excellent results [14].

The anxiolytic effect of visual, auditory and tactile distraction techniques was constantly performed by operators and technicians, as already demonstrated in other surgical fields [32], which is reinforced by the use of nitrous oxide inhalation which presents both an anxiolytic and analgesic effect [33].

Finally, the absence of general anesthesia may broaden indications to patients for which general anesthesia may be contraindicated due to a frail condition. This is probably the most important added value of local anesthesia as it has been shown that palliative radiotherapy should be performed in patients with a short life expectancy [34]. Therefore, and because pain relief is obtained immediately after procedure with RFA (as opposed to several weeks with radiation therapy), RFA under local anesthesia should evidently be considered as the method of choice in patients in the palliative care.

Our study however has several limitations. The results are retrospective based on consultation data. The study relies on a small study sample, with a lack of uniformity of the included lesions (size and type). Finally, no data were available on pain therapy modifications after the procedure, which may have introduced an outcome bias.

## 6. Conclusions

Bipolar RFA is a safe and effective treatment of painful refractory vertebral metastases in patients in palliative care. It is well tolerated under local anesthesia with nitrous oxide inhalation. This approach allows for a regular intraprocedural clinical examination and may help avoid possible surrounding neural damage, such as cord injury. Immediate pain relief is observed, improving the patient’s quality of live, which is desirable in patients with a limited life expectancy. Such results are a priority in pain palliation patients.

Further studies comparing RF alone versus RF combined with vertebroplasty are needed to establish the benefit of a combining RF and vertebroplasty.

## Figures and Tables

**Figure 1 medicina-57-00966-f001:**
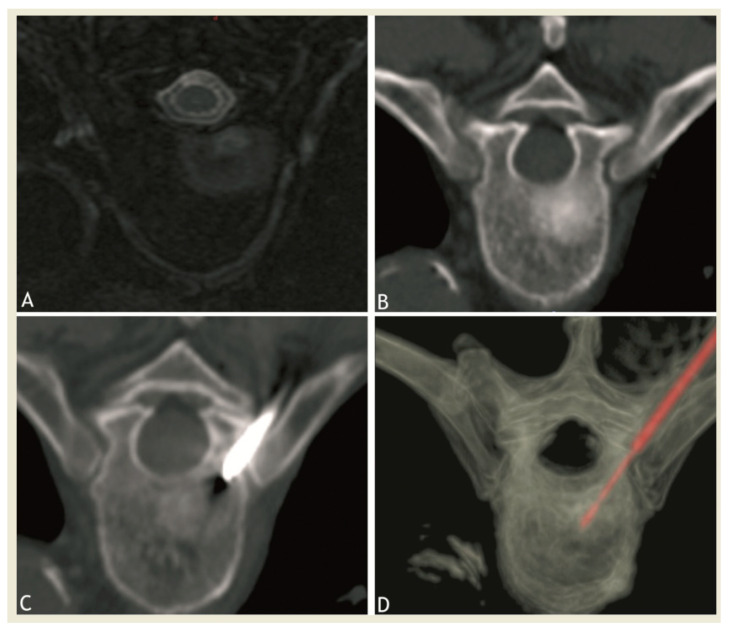
Axial MRI after gadolinium injection (**A**) and CT (**B**) showing a sclerotic-enhancing lesion of the vertebral body of fifth thoracic vertebra. (**C**,**D**): CT images showing the radiofrequency probe at target site in the lesion during thermoablation session.

**Figure 2 medicina-57-00966-f002:**
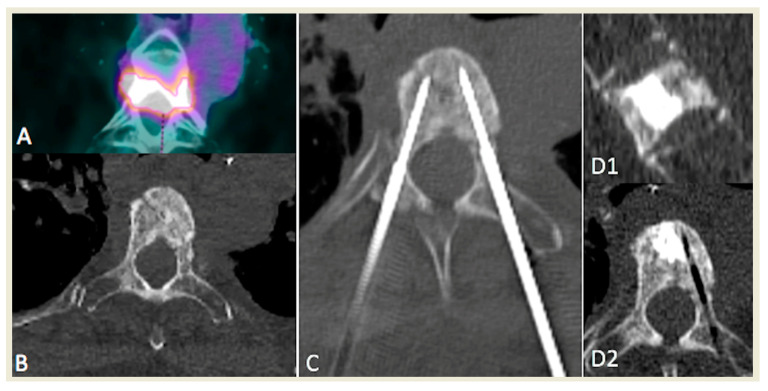
Example of a patient presenting with a unique painful, T4 hypermetabolic colorectal cancer sclerotic metastasis (**A**,**B**) treated with bipedicular bipolar radiofrequency ablation (**C**) and subsequent vertebroplasty in sagittal (**D1**) and axial (**D2**) planes.

**Figure 3 medicina-57-00966-f003:**
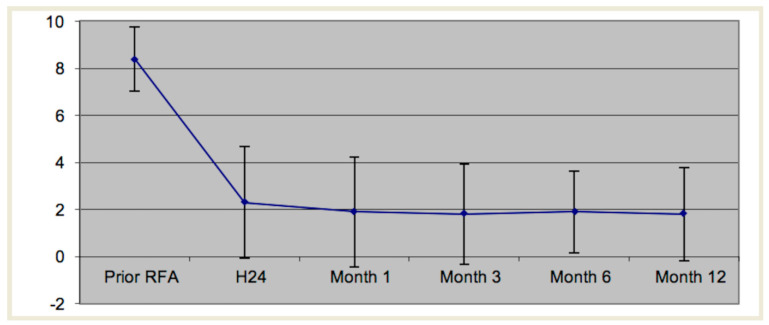
Mean visual analog scale as a function of time.

**Table 1 medicina-57-00966-t001:** Patient and lesion characteristics.

Sex/Age	Primary	Lesion Level	Size (mm)	Type	Soft-Tissue Involvement	Posterior Wall Involvement	Cementoplasty
1-M/46	melanoma	body T6	16 × 25 × 12	lytic	no	yes	yes
2-M/54	lung	body L1	29 × 35 × 13	lytic	no	no	yes
3-M/58	lung	body L2	30 × 30 × 26	lytic	no	no	yes
4-M/56	lung	body T7	15 × 26 × 13	lytic	no	no	yes
		body T8	17 × 28 × 12	lytic	no	no	yes
5-F/75	lung	body L1	61 × 58 × 20	lytic	yes	yes	yes
6-F/55	lung	body L5	25 × 12 × 33	lytic	yes	no	yes
7-M/60	lung	pedicle L T12	54 × 32 × 20	lytic	yes	no	no
8-M/72	lung	R Transverse process T5	16 × 19 × 13	lytic	yes	no	no
9-M/59	lung	body T10	32 × 35 × 20	lytic	yes	no	yes
10-M/76	prostate	body L3	32 × 27 × 19	osteoblastic	no	yes	yes
11-M/56	urothelial	body L3	13 × 13 × 12	lytic	yes	no	no
12-M/74	prostate	body T9	16 × 19 × 12	osteoblastic	no	yes	no
13-M/59	kidney	body L2	14 × 15 × 14	lytic	yes	no	yes
14-M/44	head and neck	body T12	15 × 13 × 8	osteoblastic	no	no	no
15-M/61	hepatocellular carcinoma	pedicle L L5	38 × 47 × 22	lytic	yes	yes	no
		pedicle L L4 et L5	54 × 30 × 33	lytic	yes	yes	no
16-F/57	breast	body T12	40 × 57 × 11	mixt	no	no	yes
17-M/75	pancreas	R pedicle C7/T1	55 × 44 × 40	lytic	yes	yes	no
18-M/63	lung	pedicle R L2	15 × 20 × 16	lytic	no	yes	no
19-F/69	neuroendocrine	sacrum L S1	28 × 31 × 20	lytic	no	no	no
20-M/61	pancreas	body L2	45 × 40 × 22	lytic	no	no	yes
21-F/35	breast	body L3	25 × 29 × 21	lytic	no	no	yes
22-F/46	colorectal	body L4	22 × 38 × 24	lytic	yes	no	yes
		SI right	19 × 30 × 18	lytic	no	no	yes
23-M/76	colorectal	body L1	21 × 18 × 13	lytic	no	no	yes
24-M/63	colorectal	L Transverse process T6	10 × 10 × 10	lytic	yes	no	no
25-F/66	colorectal	sacrococcygeal	56 × 55 × 71	lytic	yes	yes	no
	colorectal	sacrococcygeal	50 × 56 × 45	lytic	yes	yes	no

**Table 2 medicina-57-00966-t002:** Detailed VAS follow-up data.

Patient	VAS Scores						Tolerance
	Before Procedure	24 h	1 Month	3 Month	6 Month	12 Month	
1	10	5	0	1	2	_	1
2	10	1	2	0	0	_	0
3	9	2	1	3	4	0	0
4	9	2	2	0	0	0	1
	8	0	0	0	_	_	1
5	10	1	0	2	1	0	0
6	9	4	0	4	5	lost	0
7	9	7	0	6	3	4	0
8	10	0	9	7	_	_	1
9	10	7	3	_	_	_	1
10	9	2	0	0	4	4	1
11	8	0	0	4	2	3	1
12	8	0	0	0	0	0	0
13	9	0	3	0	0	5	0
14	6	0	0	0	0	0	0
15	8	3	4	4	0	0	1
	4	4	4	_	_	_	0
16	8	0	0	0	2	_	1
17	8	8	8	_	_	_	2
18	7	4	3	_	_	_	0
19	9	2	3	1	0	0	1
20	6	0	0	_	_	_	0
21	9	0	2	4	5	5	0
22	9	1	2	1	3	1	0
	9	2	1	2	2	2	1
23	9	4	0	0	1	_	1
24	8	4	3	3	3	3	0
25	9	1	2	0	0	0	0
	8	0	0	0	_	_	0
Mean	8.4	2.2	1.8	1.8	1.9	1.8	
Standard Deviation	±1.4	±2.4	±2.3	±2.2	±1.8	±2.1	

## Data Availability

All the data are available from the corresponding author upon reasonable request.

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
