# Peer review of "Bipolar Radiofrequency Ablation of Painful Spinal Bone Metastases Performed under Local Anesthesia: Feasibility Regarding Patient’s Experience and Pain Outcome"

_medicina, 2021, doi:10.3390/medicina57090966_

Round 1

Reviewer 1 Report

Line37-38 page 2- What do you mean by -medullary involvement? In the spinal canal?

Line 52 page 3 - management of spinal neoplasm – is rather broad, these techniques deal with the vertebrae not the spinal cord, can you make this clear?

Line 170 – looks like the sentence is incomplete and jumps to the next line.

Also please correct the citation format and make it uniform.

Author Response

Line37-38 page 2- What do you mean by -medullary involvement? In the spinal canal?

This phrase was re-written for more clarity : In case of extension to the neural structures”

Line 52 page 3 - management of spinal neoplasm – is rather broad, these techniques deal with the vertebrae not the spinal cord, can you make this clear?

The phrase ‘spinal neoplasm was replaced by “vertebral metastasis”

Line 170 – looks like the sentence is incomplete and jumps to the next line

Indeed, this has been modified

Also please correct the citation format and make it uniform.

Endnote software was used to format the references.

Reviewer 2 Report

Thanks a lot for giving me the opportunity to revise such an interesting paper Bipolar radiofrequency ablation of painful spinal bone metastases performed under local anesthesia : feasibility regarding patient’s experience and pain outcome, related to to a frequent condition like bone metastases

With a constructive intention, I would suggest some few minor points to be explane, added/modified:

The authors reported that 16 procedures were combined with vertebroplasty for bone consolidation purposes. What criteria were used for treatment with RFA alone and what criteria were used for combined thermoablation and cementoplasty treatment?

Were pathologic fractures present?

In patients with lesions near nerve structures, especially those with vertebral pedicle lesions, have thermosensors been used to measure local temperature to protect nerves during the ablation phase?

Rapid inhalation of nitrous oxide alleviates pain and anxiety promptly. However, this approach may increase discomfort in adult patients, and the probability of adverse reactions such as nausea and vomiting. Also, the use of nitrous oxide inhalation may cause other side effects such as blurred vision, confusion, dizziness, faintnessm (most common). Have you observed any side effects in your group of patients. If yes, please report it in your work.

Did patients received radiotherapy after surgery? Does this treatment avoid radiotherapy?

In line 198 nirous is nitrous?

Author Response

Thanks a lot for giving me the opportunity to revise such an interesting paper Bipolar radiofrequency ablation of painful spinal bone metastases performed under local anesthesia : feasibility regarding patient’s experience and pain outcome, related to to a frequent condition like bone metastases

With a constructive intention, I would suggest some few minor points to be explane, added/modified:

The authors reported that 16 procedures were combined with vertebroplasty for bone consolidation purposes. What criteria were used for treatment with RFA alone and what criteria were used for combined thermoablation and cementoplasty treatment?

The Kostuik score was used to determine the risk of fracture.

This has been added in the text

Were pathologic fractures present?

Yes, in 7 cases, this was added in the manuscript

In patients with lesions near nerve structures, especially those with vertebral pedicle lesions, have thermosensors been used to measure local temperature to protect nerves during the ablation phase?

Yes, thermocouple was used in 17 cases, this was added to the manuscript

Rapid inhalation of nitrous oxide alleviates pain and anxiety promptly. However, this approach may increase discomfort in adult patients, and the probability of adverse reactions such as nausea and vomiting. Also, the use of nitrous oxide inhalation may cause other side effects such as blurred vision, confusion, dizziness, faintnessm (most common). Have you observed any side effects in your group of patients. If yes, please report it in your work.

In our experience, negative experiences with nitrous oxide does occur, but these events are relatively rare especially as the way we use nitrous oxide in our patient is self administration, therefore, the patient is free to put the mask and retrieve it if discomfort is felt. Therefore, no negative experience are to be reported in our series.

Did patients received radiotherapy after surgery? Does this treatment avoid radiotherapy?

All of the included patient had already undergone radiotherapy, this was added in the material and methods paragraph.

This treatment may be an alternate in specific cases, bit to date, it must not be considered as an exclusive treatment, allowing to avoid radiotherapy, although in specific cases of small tumor, recent papers have shown that RFA allows curative treatments.

In line 198 nirous is nitrous?

This typo was corrected

Reviewer 3 Report

I'd like to thank authors for this interesting paper on spine pain relief.

I have some questions?

  • you evaluated pain but we have no information about pain killer ? Which treatment? Modification after RFA? can you provide us these informations?
  • you evaluated pain with visual scale. Did you use other questionnaries such as QOL ones ?
  • It could be very interesting for authors to have a small technical part in the discussion explaining why you used rfa and not mwa or unipolar?
  • did you have posterior lesion requiring protection (carbodissection?) or if you had posterior lesion, you just performed vertebroplasty?
  • you used parametrical test for your statistics. Did you Check gaussian distribution? How? please, provide us this information.  If not, better use non parametrical test.

I hope i could help you improve your paper.

Best regards

Author Response

I have some questions?

you evaluated pain but we have no information about pain killer ? Which treatment? Modification after RFA? can you provide us these informations?

All of the included patients had undergone Opioid therapy prior to procedure. However, modification of oral therapy after procedure was not available for assessment, which indeed may constitute a bias, this was added in the manuscript and in the limitations paragraph.

you evaluated pain with visual scale. Did you use other questionnaries such as QOL ones ?

No, unfortunately only VAS scores were available for outcome assessments in our records.

It could be very interesting for authors to have a small technical part in the discussion explaining why you used rfa and not mwa or unipolar?

A paragraph has been added in the discussion on Bipolar vs monopolar vs MWA advantages

did you have posterior lesion requiring protection (carbodissection?) or if you had posterior lesion, you just performed vertebroplasty?

Thermocouples were used to prevent neural damage. This was added in the manuscript, and follows a previous reviewers remark.

you used parametrical test for your statistics. Did you Check gaussian distribution? How? please, provide us this information.  If not, better use non parametrical test.

Shapiro–Wilk test was used to determine whether variables came from a normally distributed population.

Reviewer 4 Report

Review – Article, « Bipolar radiofrequency ablation of painful spinal bone metastases performed under local anesthesia: feasibility regarding patient’s experience and pain outcome »

Summary:

The authors have performed a retrospective study evaluating the feasibility and tolerance of bipolar RFA combined or not with vertebroplasty in painful vertebral metastases performed under local anesthesia. The authors found that bipolar RFA with or without vertebroplasty is feasible under local anesthesia and nitrous oxide inhalation with a good tolerance leading to a decrease of VAS score up to 12 months after the procedure.

Comments:

  1. Introduction: In addition to the 20% of patients who are not responders to radiation therapy and to the delayed effect, considered the additional value of percutaneous bone ablation which is repeatable, and which has not limitation doses from the skin exposure. Moreover, it might be interesting to consider the absence of interference with systemic treatments especially those delaying healing (contrary to the open surgery procedure).
  1. Material and methods: It might be helpful to specify analgesic consumption of patients included in the study as it can interfere with the basal pain level before percutaneous ablation. Then, it might be useful to explicit that any patient presenting with unstable pathologic vertebral fracture (SINS >7) or metastatic epidural cord compression were included. It might be interesting to explicit if percutaneous ablation is decided during a multidisciplinary meeting.
  1. RFA technique: The authors report, “Multiple RFA cycles and/or multiple needle approaches…for large lesions (> 4 cm). It might also be examined if the localization of the lesion (either vertebral body, pedicle, sacral lesion) or an extent to the soft tissue modified the needle choice. In addition, the definition of a technical success when using multiple RFA probes should be explained especially the distance between probes.
  2. Pain assessment: Regarding the impact of pain on quality of life, a scale describing the improvement of quality of life could be provided (in example: Oswestry disability index)

  3. Procedure: It might be helpful to further underline the value of neurological assessment during the procedure under local anesthesia. Indeed, permanent communication between the physician and the patient, repeatable neurologic examination and immediate monitoring of neurological disorder can be done. Moreover, the lack of anesthesia drugs inhibiting the neural function is also a great advantage. Late, it might be precise if any ancillary protective technique (as hydric or gas displacement) were used especially for sensitive localization.

General overview:

In general, the manuscript is well-written demonstrating the feasibility and the tolerance of this effective procedure under local anesthesia performed by interventional radiologists on pain in a one-year follow-up period.

Author Response

Comments:

  1. Introduction: In addition to the 20% of patients who are not responders to radiation therapy and to the delayed effect, considered the additional value of percutaneous bone ablation which is repeatable, and which has not limitation doses from the skin exposure. Moreover, it might be interesting to consider the absence of interference with systemic treatments especially those delaying healing (contrary to the open surgery procedure).

Thank you for this comment which has been added in the introduction

  • Material and methods: It might be helpful to specify analgesic consumption of patients included in the study as it can interfere with the basal pain level before percutaneous ablation.

Unfortunateley, we do not have the details of the oral therapy modification secondary to the procedure. All included patients presented refractory pan to opoid therapy, but no info in our records allowed to assess possible changes in the therapy after procedure. This was added in the limitations paragraph

  • Then, it might be useful to explicit that any patient presenting with unstable pathologic vertebral fracture (SINS >7) or metastatic epidural cord compression were included. It might be interesting to explicit if percutaneous ablation is decided during a multidisciplinary meeting.

We used the Kostuik scoring system, for stability assessment. This was added in the manuscript

  1. RFA technique: The authors report, “Multiple RFA cycles and/or multiple needle approaches…for large lesions (> 4 cm). It might also be examined if the localization of the lesion (either vertebral body, pedicle, sacral lesion) or an extent to the soft tissue modified the needle choice. In addition, the definition of a technical success when using multiple RFA probes should be explained especially the distance between probes.

We have added the minimal distance between 2 probes which 1.5 cm in the technical success

  1. Pain assessment: Regarding the impact of pain on quality of life, a scale describing the improvement of quality of life could be provided (in example: Oswestry disability index)

Indeed, this would be interesting to assess, unfortunately no scale on quality of life was available in our record for assessment

  1. Procedure: It might be helpful to further underline the value of neurological assessment during the procedure under local anesthesia. Indeed, permanent communication between the physician and the patient, repeatable neurologic examination and immediate monitoring of neurological disorder can be done. Moreover, the lack of anesthesia drugs inhibiting the neural function is also a great advantage. Late, it might be precise if any ancillary protective technique (as hydric or gas displacement) were used especially for sensitive localization.

Details on the added value of clinical neurological assessment during procedure were added, and information on thermocouple  use was also added to the manuscript